# Effects of High Heat Load Conditions on Blood Constituent Concentrations in Dorper, Katahdin, and St. Croix Sheep from Different Regions of the USA

**DOI:** 10.3390/ani12172273

**Published:** 2022-09-02

**Authors:** Dereje Tadesse, Amlan Kumar Patra, Ryszard Puchala, Arthur Louis Goetsch

**Affiliations:** 1American Institute for Goat Research, Langston University, Langston, OK 73050, USA; 2Department of Animal Sciences, Debre Berhan University, Debre Berhan P.O. Box 445, Ethiopia; 3Department of Animal Nutrition, West Bengal University of Animal and Fishery Sciences, Kolkata 700037, India

**Keywords:** blood constituents, breed difference, hair sheep, heat stress

## Abstract

**Simple Summary:**

High heat load (HL) can adversely affect livestock health and cause significant production losses as a result of alterations of many physiological functions including nutrient metabolism, oxidative status, blood constituent levels, and gut dysbiosis. Effects of high HL could vary with breed of hair sheep (i.e., Dorper, Katahdin, and St. Croix) as well as region of the United States of America with different climatic conditions where they originated. Differences among the three sheep breeds in levels of a number of blood constituents were observed. Different HL conditions also altered concentrations of several blood metabolites and constituents, including hemoglobin, oxygen concentration, creatinine, heat shock protein, thyroxine, and cortisol. However, there were no interactions between breed and HL for most blood variables, suggesting similar resilience to high HL. In addition, only a few differences among or interactions involving region suggest that the breeds are highly adapted to diverse climatic conditions.

**Abstract:**

Forty-six Dorper (DOR), 46 Katahdin (KAT), and 43 St. Croix (STC) female sheep (initial body weight of 58, 59, and 46 kg, respectively, SEM = 1.75; 3.3 ± 0.18 years of age, 2.6–3.7), derived from 45 commercial farms in four regions of the USA (Midwest, Northwest, Southeast, and central Texas), were used to evaluate responses in blood constituent concentrations to increasing heat load index (HLI) conditions. There were four sequential 2 weeks periods with target HLI during day/nighttime of 70/70 (thermoneutral zone conditions), 85/70, 90/77, and 95/81 in period 1, 2, 3 and 4, respectively. A 50% concentrate pelletized diet was fed at 53.3 g dry matter/kg body weight^0.75^. The analysis of most constituents was for samples collected on the last day of the second week of each period at 13:00 h; samples for cortisol, thyroxine, and heat shock protein were collected in week 2 and 8. Previously, it was noted that resilience to high HLI conditions was greatest for STC, lowest for DOR, and intermediate for KAT. There were few effects of region. Other than hemoglobin concentration, there were no interactions between breed and period. Blood oxygen concentration was greatest (*p* < 0.05) among breeds for STC (5.07, 5.20, and 5.53 mmol/L for DOR, KAT, and STC, respectively; SEM = 0.114) and differed among periods (4.92, 5.26, 5.36, and 5.52 mmol/L for period 1, 2, 3, and 4, respectively; SEM = 0.093). There were breed differences (i.e., main effects; *p* < 0.05) in glucose (50.0, 52.6, and 52.1 mg/dL; SEM = 0.76), urea nitrogen (17.2, 17.3, and 19.4 mg/dL; SEM = 0.33), creatinine (0.991, 0.862, and 0.802 mg/dL; SEM = 0.0151), total protein (6.50, 6.68, and 6.95 g/l; SEM = 0.017), triglycerides (28.4, 29.1, and 23.5 mg/dL; SEM = 0.87), and cortisol (6.30, 8.79, and 6.22 ng/mL for DOR, KAT, and STC, respectively; SEM = 0.596). Differences among periods (*p* < 0.05) were observed for lactate (27.9, 25.3, 27.8, and 24.0 mg/dL; SEM = 0.99), creatinine (0.839, 0.913, 0.871, and 0.917 mg/dL; SEM = 0.0128), total protein (6.94, 6.66, 6.60, and 6.65 g/l; SEM = 0.094), and cholesterol (60.2, 56.5, 58.3, and 57.6 mg/dL for period 1, 2, 3, and 4, respectively; SEM = 1.26). In addition, the concentration of cortisol (7.62 and 6.59 ng/mL; SEM = 0.404), thyroxine (5.83 and 5.00 µg/dL; SEM = 0.140), and heat shock protein (136 and 146 ng/mL for week 2 and 8, respectively; SEM = 4.0) differed between weeks (*p* < 0.05). In conclusion, the lack of interaction between breed and period with different HLI conditions suggests that levels of these blood constituents were not highly related to resilience to high HLI.

## 1. Introduction

Production of hair sheep for meat has been increasing in the United States of America and other countries over many years [1,2,3,4]. Although US Government statistics officially do not maintain breed inventory, breed association registration records indicate that Dorper and Katahdin registrations have increased by over 8000 and 10,000 in 2017 from around 2000 and 4000 in 2000, respectively [5]. The most important hair sheep breeds in the USA currently are Dorper, Katahdin, and St. Croix. Each breed has a unique history in terms of original development, such as geographic location, base breeds, traits emphasized in selection, etc. [4]. Therefore, naturally, the breeds vary in many characteristics, including size, conformation, body composition, resistance to common stress factors, growth, reproduction, and rate and pattern of maturation [6]. In addition, now with these breeds on farms in various areas of the USA over numerous years, it is likely that such characteristics have changed to some extent. Hence, for most appropriate comparisons, it would be desirable to include animals from multiple sources, considering conditions such as regions with varying climatic conditions. In this regard, recently, studies have been conducted to evaluate responses of these hair sheep breeds to various stress factors that are expected to increase in importance with anticipated climate change, namely limited feed intake [7], restricted availability of drinking water [8], and high heat load conditions [9]. Heat stress may be the stress factor related to climate change most significant in regard to future livestock production. Heat stress alters many physiological mechanisms such as metabolism and immune function, oxidative status, reproductive processes, blood electrolyte balance, and imbalance of the gut microbiota and barrier function, which all can affect animal performance, health, and welfare and economic losses [10,11,12,13]. In this regard, livestock reared in extensive systems may experience periodic heat waves and stressful environmental conditions at some times of the year, particularly in tropical and subtropical regions [13,14,15]. Seasonal heat stress can have dramatic detrimental impacts on animal production and reproduction [10,15,16].

Various means have been used to assess conditions that elicit heat stress in livestock. Common ones are temperature–humidity index (THI; [17]) and heat load index (HLI). Temperature and humidity have major influence on both THI and HLI [18,19]. However, other factors can have impact on heat stress as well, such as solar radiation and wind [19,20,21]. Even though animals in this study were inside a confinement facility and, thus, not exposed to direct sunlight or wind, it was felt desirable to employ a measure of conditions causing heat stress that would be most appropriate for a wide array of environmental settings. Hence, the HLI of Gaughan et al. [21] was employed, which considers black globe temperature, relative humidity, and wind speed. Although, the THI of Amundson et al. [17] is also presented by Tadesse et al. [9] in a report on other data of this study.

Tadesse et al. [9] subjected Dorper, Katahdin, and St. Croix sheep derived from four regions of the USA to increasing HLI conditions and reported on variables such as body weight, feed intake, rectal and skin temperature, respiration rate, and panting score. Based primarily on rectal temperature and respiration rate, it was concluded that resilience to high heat load index conditions ranked St. Croix > Katahdin > Dorper, with lowest variability among individuals for St. Croix and greatest for Katahdin. These findings presumably relate to different phenotypic and morphological characteristics including skin and coat color, coat type, fat deposition, sweat gland number, and body conformation that influence responses to high HLI in functions such as catabolic and anabolic hormones, metabolic rates, nutrient partitioning, and molecular and gene modifications, which could be reflected in the blood biochemical profile [22,23]. Hence, levels of some hemato-biochemical constituents could be important response indicators of thermal challenge in sheep [14]. Therefore, objectives of this report are to address how concentrations of some blood constituents of these different hair sheep breeds from various regions might have been affected by increasing HLI.

## 2. Materials and Methods

### 2.1. Animals, Housing, and Diet

Many conditions of the study have been described by Tadesse et al. [9], which are briefly overviewed. Protocols for trials of the study were approved by the Langston University Animal Care and Use Committee. The experiment consisted of four trials and animal sets, conducted in the fall of 2015, spring and fall of 2016, and spring of 2017. Forty-six Dorper (DOR), 46 Katahdin (KAT), and 43 St. Croix (STC) female sheep from 45 commercial farms were used. They were obtained in the summer of 2015 from four regions of the USA with different climatic conditions, representing ’ecotypes.’ Regions were the Midwest (MW; portions of Iowa, Minnesota, Wisconsin, and Illinois), Northwest (NW; primarily Oregon with one farm in southern Washington and another near Seattle), Southeast (SE; Florida and one farm in southern Georgia), and central Texas (TX). The design of the experiment is overviewed in Figure 1.

Most animals were ewes when procured, although a small number were lambs. Age at the start of the trials averaged 3.3 ± 0.18 yr, ranging from 2.6 to 3.7. Initial body weight (BW) was 57.9, 58.9, and 45.6 kg (SEM = 1.75) for DOR, KAT, and STC, respectively. Procedures employed were largely based on the study of Mengistu et al. [24]. Animals were housed in one room individually in elevated pens with a plastic-coated expanded metal floor at most times. A 50% concentrate pelletized diet was fed at 53.3 g dry matter (DM)/kg BW^0.75^, which was approximately 120% of an assumed metabolizable energy requirement. Diet composition was described by Tadesse et al. [9]. Feed was offered twice daily at 08:00 and 15:00 h. In addition, 10 g/day of a supplement consisting of 95% soybean meal and 5% of a vitamin premix to provide approximately 4405, 881, and 0.55 IU/day of vitamins A, D_3_, and E, respectively, was mixed with the morning feed. Animals had free access to small pieces of trace mineralized salt blocks (96.5–99.5% NaCl, 4000 mg/kg Zn, 1600 mg/kg Fe, 1200 mg/kg Mn, 260–390 mg/kg Cu, 100 mg/kg I, and 40 mg/kg Co) placed in feed barrels and fresh or tap water.

The entire study period was 10 wk, with the first week for adaptation to the facility and thermoneutral conditions and the last week for readjustment to thermoneutral conditions. Animal measurements occurred in the 8 weeks experiment, with four 2 weeks periods. Heat load index (HLI) was calculated as proposed by Gaughan et al. [21] for temperatures above 25 °C: 8.62  +  0.38 × RH  +  1.55 × BGT – 0.5 × WV  +  e(2.4 – WV), where RH = relative humidity (decimal form), BGT = black globe temperature (°C), WS = wind velocity (m/s) and e is the base of the natural logarithm. Target HLI during the daytime was of 70, 85, 90, and 95 and that in nighttime was 70, 70, 77, and 81 in periods 1, 2, 3, and 4, respectively. In higher HLI periods, the level of HLI during nighttime was approximately 85% of HLI during daytime, similar to that of Hamzaoui et al. [25]. Daytime conditions were between 07:00 and 19:00 h. As noted by Tadesse et al. [9], HLI values in the periods were fairly close to those intended. Although, the values generally were slightly greater than targets, more so in the night than in daytime and in period 2 than at other times.

### 2.2. Measurements

Blood samples were collected via jugular venipuncture using 10 mL tubes with and without sodium heparin every week at 13:00 h on Friday, the last day of the weeks. Immediately after sampling, hemoglobin (Hb) concentration and oxygen saturation were determined in heparinized blood with a OSM 3 Hemoximeter™ (Radiometer, Westlake, OH, USA); blood oxygen concentration was calculated as described by Eisemann and Nienaber [26]. Packed cell volume (PCV) was determined with heparinized tubes (Clay Adams, Parsippany, NJ, USA). Serum was obtained by centrifugation for 15 min at 3000 × *g* and frozen at −20 °C. Thawed serum collected in week 2, 4, 6, 7, and 8 was analyzed for lactate (LAC), albumin (ALB), urea nitrogen (UN), cholesterol (CHL), creatinine (CRT), glucose (GLC), total protein (TP), triglycerides (TG), and thyroxine (THY) with a Vet Axcel^®^ Chemistry Analyzer (Alfa Wassermann Diagnostic Technologies, West Caldwell, NJ, USA) according to the manufacturer instructions. Concentrations of cortisol (COR) and heat shock protein 70 (HSP) were determined in samples from week 2 and 8 with ELISA kits of Enzo Life Sciences (Farmingdale, NY, USA) and Wuhan Fine Biotech Co., Ltd. (Wuhan, China), respectively.

### 2.3. Statistical Analysis

Data were analyzed using a mixed effects model with SAS, version 9.3 [27,28]. The first model for variables measured each week (i.e., Hb concentration and O_2_ saturation, O_2_ concentration in blood, and PCV) included fixed effects of animal set, initial age as a covariate, breed, region, period, week within period, and all interactions other than ones involving animal set and age. Animal within breed × region was random. Because there were many significant effects of week within period and interactions involving it as shown in Table 1, an analysis was conducted with measures in the first week of each period omitted under the assumption of more stable physiological conditions in the second week. The same model was used for concentrations of other blood constituents determined in the second week of periods. Moreover, to evaluate adaptation with the highest HLI conditions, these variables measured in both weeks of period 4 were analyzed with the repeated measure of week. Means were separated by least significant difference when the treatment F-test was significant (*p* < 0.05).

## 3. Results

### 3.1. Hb, Oxygen, and PCV—All Periods

There were interactions (*p* < 0.05) involving period and breed in Hb concentration and region in oxygen saturation of Hb (Table 2). For Hb concentration, this was mainly because of less change with advancing period for STC than for DOR and KAT (Table 3). Oxygen saturation of Hb was low in period 1 for NW and high in period 3 for TX relative to other regions. Despite these interactions, blood oxygen concentration differed (*p* < 0.05) only among breeds and periods, with the highest value for STC and a period ranking of 4 > 2 > 1 (*p* < 0.05), with the value for period 3 not different from those in periods 2 and 4. The PCV was greater for MW than for NW and SE and greatest among periods for period 1 (*p* < 0.05).

### 3.2. Blood Chemistry—All Periods

There were no significant interactions involving period in concentrations of these 11 blood constituents (Table 4). Period affected (*p* < 0.05) levels of seven of the variables, LAC, CRT, TP, CHL, COR, THY, and HSP, and tended to have an effect on ALB (*p* = 0.067). The concentration of LAC was lower (*p* < 0.05) in period 4 than in 1 and 3, with an intermediate value for period 2 (*p* > 0.05). The TP concentration was highest among periods (*p* < 0.05) for period 1, and there was a similar numerical difference for ALB. The concentration of CHL was higher (*p* < 0.05) for period 1 vs. 2 and 4, with an intermediate value for period 3 (*p* > 0.05). Levels of COR and THY were greater in period 1 vs. 4, whereas that of HSP was higher in period 4 (*p* < 0.05).

Breed affected six of the eleven variables (*p* < 0.05), GLC, UN, CRT, TP, TG, and COR, although there was a breed × region interaction for UN and TG (*p* < 0.05; Table 4). The level of GLC was lowest among breeds for DOR (*p* < 0.05; Table 5). Overall, UN was greatest among breeds for STC (*p* < 0.05), although values for TX were similar. The level of CRT ranked (*p* < 0.05) DOR > KAT > STC. The TP concentration was greater for STC vs. DOR (*p* < 0.05), with an intermediate value for KAT (*p* > 0.05). Overall, the TG level was lower for STC than for DOR and KAT (*p* < 0.05), but the greatest difference between STC and KAT was for NW and that between STC and DOR was for TX. The concentration of COR was considerably greater for KAT vs. DOR and STC (*p* < 0.05). The concentration of LAC was greater for MW and NW than for TX (*p* < 0.05), with an intermediate value for SE (*p* > 0.05).

### 3.3. Blood Chemistry—Week of Period 4

The Hb concentration differed (*p* < 0.05) between week 7 and 8 for KAT (i.e., lower in week 8) but not for DOR or STC (Table 6 and Table 7). The only other breed × week interaction was for PCV, although values were lower in week 8 for both KAT and DOR (*p* < 0.05). There was one variable affected by an interaction between region and week, which was CRT. Values for MW and TX were greater in week 8 vs. 7 (*p* < 0.05), in contrast to similar values for NW and SE. The concentration of LAC was less in week 8 vs. 7 (*p* < 0.05). There were no variables for which the main effect of region was significant. There were six variables affected by breed, with a significant breed × region interaction for two (*p* < 0.05). Oxygen saturation of Hb was lowest among breeds for DOR (*p* < 0.05), although blood oxygen concentration was similar among breeds. The UN concentration was greater for STC than for other breeds, but values for TX were similar among breeds. The CRT concentration was highest among breeds for DOR (*p* < 0.05). The region × week interaction in CRT concentration involved greater levels for week 8 vs. 7 for MW and TX (*p* < 0.05) and similar values for NW and SE. The TP concentration was greater for STC than for DOR and KAT (*p* < 0.05), and the level of ALB was greater for STC vs. DOR (*p* < 0.05) but not for KAT. The level of TG was lowest among breeds for STC (*p* < 0.05), but differences varied markedly among regions. The concentration of CHL was not affected by breed, region, week, or any interaction (*p* > 0.05).

## 4. Discussion

### 4.1. Hb, Oxygen, and PCV—All Periods

Decreases in Hb concentration and PCV from period 1 to 4 of approximately 1.3 g/dL and 2.8 percentage units, respectively, are in accordance with findings of some previous studies [24,29,30]. Although water intake was not recorded in the present experiment, from observations, it appeared to increase markedly with increasing HLI as the period advanced, which may have contributed to these changes [31]. However, this assumes that water intake was greater than required for the increasing need for use in evaporative cooling. In this regard, in other studies, heat stress has increased Hb concentration, with dehydration as the proposed reason [32,33]. Another factor regarding Hb concentration is oxidative stress in response to high HLI that can lead to denaturing and precipitation of Hb in erythrocytes, followed by their degradation [34,35]. However, magnitudes of change in Hb and PCV were relatively small, and as noted below, change in oxygen saturation more than compensated for the effect of period on Hb.

The marked increase in respiration rate as a period advanced and HLI increased [9] presumably was responsible for increased oxygen saturation of Hb, which again facilitated a general increase in blood oxygen concentration with advancing period. Breed rankings in oxygen saturation of Hb and blood oxygen concentration are opposite those noted in respiration rate [9]. With the ranking in resilience to high HLI of STC > KAT > DOR of Tadesse et al. [9], this suggests that breaths of STC were the deepest (i.e., greatest volume of air being moved) relative to less deep or more shallow respiration of KAT and, in particular, DOR. Animals with a rapid and shallow breathing pattern have been reported to have difficulty in saturating blood with oxygen because of less carbon dioxide removal [36].

### 4.2. Blood Chemistry—All Periods

#### 4.2.1. Period

An important finding of this experiment is the lack of interaction in blood chemistry variables between period and breed and region. This indicates that the effects of increasing HLI as the period advanced, when they occurred, were similar for the three breeds of hair sheep and the four regions of the USA from where they originated. This seems quite notable given the many variables affected by period and breed and the fact that, based on findings of Tadesse et al. [9], resilience to high HLI varied considerably among breeds. Therefore, physiological conditions responsible for breed differences in blood constituent concentrations evidently were affected by HLI in similar manners. As noted by Tadesse et al. [9], no, few, or minor effects of region could involve some adaptation to conditions of the study site given the time required for conduct of the trials after animals were obtained.

As generalized by Tadesse et al. [37], the blood constituents GLC, LAC, TG, and CHL relate to energy supply and demand, and UN, CRT, TP, ALB, and COR provide information about protein status. Thyroxine concentration and rate of energy metabolism are related, and HSP may reflect degree of heat stress.

The lack of differences or marked ones among periods in blood constituent concentrations related to energy status would in part relate to no or small differences among periods in DM intake in kg/day. Although, as noted for PCV and Hb concentration, increased water intake and body water content could have affected blood constituent concentrations. As GLC and LAC metabolism are highly related, it is unclear why GLC concentration was similar among periods and that of LAC in period 4 was relatively low. Lactate is produced in response to an insufficient oxygen supply in skeletal muscle, liver, and other organs, and its concentration in blood increases if the rate of production is greater than that of oxidation in the liver [38]. In the present study, an oxygen limitation in period 4 would not appear to have been a factor affecting LAC production rate given the general increase in blood oxygen concentration with increasing HLI and advancing period. The level of TG was similar among periods, although the concentration of CHL was higher in period 1 vs. 2 and 4. Likewise, higher levels of CHL in sheep and goats have been noted with the higher seasonal temperature in summer than spring [39].

The lower level of THY in period 4 vs. 1 suggests a lower rate of energy metabolism with highest HLI conditions. Heat stress can affect activity of the hypothalamus–pituitary–thyroid axis [23]. Thyroxin increases oxygen consumption and heat production in body cells, thus stimulating basal metabolic rate by increasing GLC available to cells, protein synthesis, and lipid metabolism [40,41]. Thus, the observed difference in THY would be considered a physiological adaptation to the change in environmental conditions to lower heat production and regulate heat balance [14,40,42]. This result is consistent with those found with other sheep breeds subjected to high heat load conditions [22,43,44].

Blood CRT concentration is an indicator of muscle mass. The lowest concentration of CRT among periods in period 1 may reflect minimal protein catabolism with thermoneutral conditions. Similarly, thermal challenges to dairy ewes [45] and goats [46] increased blood CRT concentration, suggesting that heat stress causes muscle degradation. That the level of TP was lower in period 4 vs. 1 despite this difference in CRT could involve relatively high water intake and body water content in period 4. Therefore, it seems that lower TP concentrations in period 4 could relate to decreased protein synthesis with heat stress [23]. Decreased protein synthesis is supported by a lower level of THY in period 1 than 4. Lower concentrations of TP in blood were also noted in heat-stressed dairy cows [47].

Cortisol, a major glucocorticoid, is mainly produced in the adrenal cortex, which is regulated by hypothalamus–hypophyseal–adrenal axis activity through corticotropin-releasing hormone from the hypothalamus and adrenocorticotropic hormone from the hypophysis. It is considered an important marker of stress and participates in various body functions including immune responses and metabolism of protein, carbohydrate, and fat [13,48]. Although high HLI conditions would be expected to elicit stress, there are findings suggesting that the lower level of COR in period 4 versus 1 might be attributable to continuous stimulation of the adrenal cortex for a long period of time, which eventually led to decreased COR synthesis. In one study, the COR concentration in plasma of sheep exposed to seasonal heat loads during summer months was decreased [49]. In a study with mature non-lactating Jersey cows, the COR concentration increased from 30 to 45 µg/L after the exposure to acute heat stress of a moderate level (35 °C) for 4 h; however, after 7 to 10 weeks of exposure to the same conditions, the COR concentration decreased to 30 to 25 µg/L [50]. There were no significant changes in COR concentration during the first 4 h of return to thermoneutral conditions (18 °C), but over the next 9 days, the COR level progressively returned to the pre-exposure level of 30 µg/L [50]. The reduction in COR concentration with long-term heat stress is beneficial to the body by lowering the metabolic rate and heat production, thus minimizing or preventing tissue damage [22].

#### 4.2.2. Breed

The higher level of HSP in period 4 vs. 1 infers that, although adaptation in physiological processes occurred, changes were insufficient to prevent some level of stress as a result of high HLI conditions [9], despite physiological adaptations such as reduced THY and COR concentrations. Heat shock proteins are stress proteins and molecular chaperones that protect the internal cell environment by participating in protein folding, repair, degradation, and many essential processes, thus regulating cell growth and survival [12,51]. Various biotic and abiotic stressors including heat stress, oxidative stress, infections, diseases, and hypoxia stimulate the production of HSP for protecting cell proteins from oxidative damage and other cellular injuries [12,52].

In the study of Tadesse et al. [37], with two levels of feed intake, one near the maintenance requirement without feed restriction and a second 55% of the initial one, GLC concentration was similar among breeds. In contrast, in the present experiment GLC concentration for DOR was lowest among breeds. Factors responsible for this difference are unclear, although the magnitude was relatively small in accordance with a similar LAC concentration among breeds. The lower level of TG for STC vs. DOR and KAT is in agreement with findings of Tadesse et al. [37], with the level based on all samples collected for STC numerically lower (28.4, 28.5, and 26.2 mg/dL for DOR, KAT, and STC, respectively; SEM = 0.84) and that determined from concentrations in the last week of the maintenance and restriction phases significantly lowest for STC (29.8, 29.5, and 26.7 mg/dL for DOR, KAT, and STC, respectively; SEM = 0.88). As discussed by Tadesse et al. [9], body condition score was lowest for STC, which could have minimized lipid mobilization, and there should have been less need for energy given the lower respiration rate of STC and greater resilience to or tolerance of high HLI by STC. Moreover, DM intake was greatest among breeds for STC (1.89, 1.88, and 2.01% BW for DOR, KAT, and STC, respectively; SEM = 0.024). No difference among breeds in CHL is in agreement with findings of Tadesse et al. [37].

Concentrations of UN and CRT are in accordance with those of Tadesse et al. [37], with the highest level of UN and lowest CRT among breeds for STC. However, in the current experiment, CRT concentration was greater for DOR than for KAT, which was not observed by Tadesse et al. [37]. The level of TP in the present study was highest for STC, but no breed differences were observed by Tadesse et al. [37]. Tadesse et al. [37] explained the differences in UN and CRT for STC based primarily on the lower body condition score [9], although higher feed intake relative to BW may have been involved as well. Tadesse et al. [37] also suggested that differences among breeds in CRT could have been due to ones in muscle mass, being presumably greatest for DOR and lowest for STC. Reasons for the highest concentration of COR among breeds for KAT are not readily apparent, and Tadesse et al. [37] did not observe a breed difference in COR.

### 4.3. Blood Chemistry—Week of Period 4

Overall, no significant three-way interactions and only two two-way interactions in concentrations after 1 and 2 weeks of period 4 indicate that rates at which physiological conditions changed with advancing time of subjection to the highest HLI conditions did not appreciably vary among breeds or regions. Relatedly, breed × week interactions in Hb concentration and PCV were due primarily to relatively low values for KAT in the second week (i.e., week 8). The region × week interaction in CRT was due mainly to a low value for MW in week 7. Reasons for the difference between week 7 and 8 in LAC are not readily apparent. Heat stress increases blood LAC concentrations due to blood acid-balance imbalance [53,54] and the reduction of LAC level in week 8 versus 7 at the greatest HLI might indicate better physiological adaptation at week 8. Nonetheless, it would appear that the majority of adaptation in physiological processes in response to the final increase in HLI occurred within the first week of the 2 weeks period, with relatively little change thereafter.

## 5. Summary and Conclusions

There were numerous differences in blood constituent concentrations among the three hair sheep breeds, Dorper, Katahdin, and St. Croix. Different heat stress conditions in the periods altered the concentration of a number of blood metabolites and constituents, including hemoglobin, oxygen, creatinine, cholesterol, total protein, heat shock protein 70, thyroxine, and cortisol. Previously, it was noted that resilience to high heat load index conditions was greatest for St. Croix, lowest for Dorper, and intermediate for Katahdin based primarily on physiological variables of rectal temperature and respiration rate. However, because of no interactions between breed and the four periods with different heat load index conditions, differences among breeds in these blood variables did not seem involved in or predictive of resilience to high heat load index, unlike physiological variables. Likewise, there were few differences among or interactions involving region where animals originated from.

## Figures and Tables

**Figure 1 animals-12-02273-f001:**
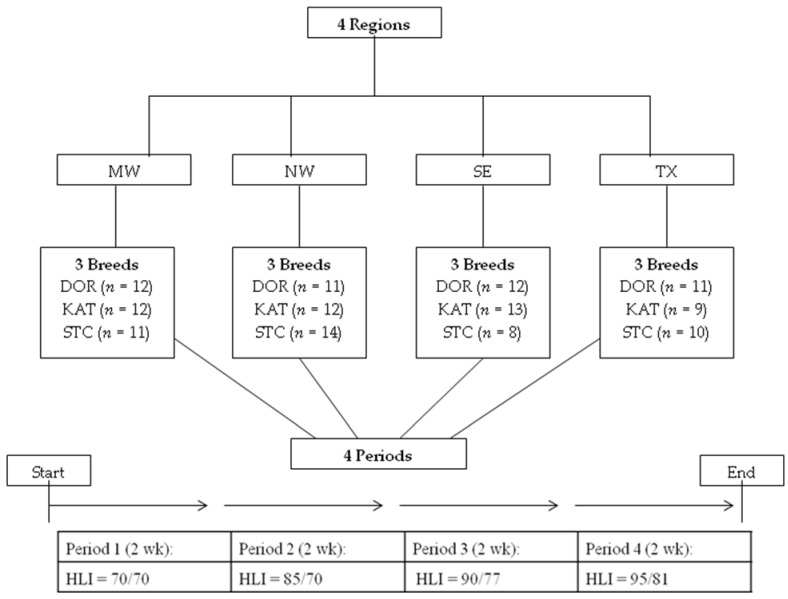
Schematic presentation of the experimental design to evaluate effects of different heat load index (HLI) conditions on three breeds, namely, Dorper (DOR), Katahdin (KAT), and St. Croix (STC) from four different climatic regions, i.e., Midwest (MW), Northwest (NE), Southeast (SE), and central Texas (TX), of USA.

**Table 1 animals-12-02273-t001:** *p* values for effects of region, breed, and period with different heat load indexes, and week within period on hemoglobin concentration and oxygen saturation, blood oxygen concentration, and packed cell volume in hair sheep based on samples collected at the end of each week of the four 2 weeks periods ^1^.

	Variable ^2^			
Source of Variation	Hb (g/dL)	HbO_2_ (%)	O_2_ (mmol/L)	PCV (%)
Set	0.060	<0.001	<0.001	0.010
Age	0.627	0.246	0.214	0.611
Breed	0.840	0.001	0.013	0.350
Region	0.140	0.633	0.279	0.085
Breed × region	0.297	0.204	0.122	0.435
Period	<0.001	<0.001	<0.001	<0.001
Breed × period	0.452	0.824	0.785	0.604
Region × period	0.917	0.273	0.681	0.901
Breed × region × period	0.539	0.687	0.454	0.587
Week	0.004	0.007	0.064	0.020
Breed × week	0.541	0.918	0.716	0.239
Region × week	0.501	0.510	0.732	0.070
Period × week	<0.001	<0.001	0.023	<0.001
Breed × region × week	0.527	0.593	0.770	0.467
Breed × period × week	0.018	0.913	0.889	0.026
Region × period × week	0.770	0.734	0.381	0.665
Breed × region × period × week	0.278	0.974	0.924	0.536

^1^ Samples were collected at 13:00 h at the end of each week of the four 2 weeks periods with increasing heat load index. Target daytime/nighttime heat load indexes were 70/70, 85/70, 90/77, and 95/81 in period 1, 2, 3, and 4, respectively. ^2^ Hb, hemoglobin; O_2_, oxygen; PCV, packed cell volume.

**Table 2 animals-12-02273-t002:** *p* values for effects of region, breed, and period with different heat load indexes on hemoglobin concentration and oxygen saturation, blood oxygen concentration, and packed cell volume (PCV) in hair sheep based on samples collected at the end of the week 2 of each of the four 2 weeks periods ^1^.

	Variable ^2^			
Source of Variation	Hb (g/dL)	Hb O_2_ (%)	O_2_ (mmol/L)	PCV (%)
Set	0.012	<0.001	<0.001	0.003
Age	0.714	0.374	0.335	0.472
Breed	0.702	0.004	0.021	0.177
Region	0.103	0.338	0.192	0.035
Breed × region	0.255	0.223	0.136	0.422
Period	<0.001	<0.001	<0.001	<0.001
Breed × period	0.019	0.703	0.771	0.062
Region × period	0.991	0.047	0.085	0.967
Breed × region × period	0.154	0.941	0.654	0.167

^1^ Samples were collected at 13:00 h at the end of week 2, 4, 6, and 8, which were the second weeks in each 2 weeks period with increasing heat load index. Target daytime/nighttime heat load indexes were 70/70, 85/70, 90/77, and 95/81 in period 1, 2, 3, and 4, respectively. ^2^ Hb, hemoglobin; O_2_, oxygen; PCV, packed cell volume.

**Table 3 animals-12-02273-t003:** Effects of breed, region, and period with different heat load indexes on hemoglobin concentration and oxygen saturation, blood oxygen concentration, and packed cell volume (PCV) in hair sheep based on samples collected at the end of the week 2 of each of the four 2 weeks periods ^1^.

		Breed ^2^				Region ^3^					Period ^4^				
Item ^5^	Period	DOR	KAT	STC	SEM	MW	NW	SE	TX	SEM	1	2	3	4	SEM
Hb (g/dL)						12.5	12.0	12.0	12.2	0.19					
	1	12.8 ^efg^	13.0 ^g^	12.9 ^fg^	0.20										
	2	12.1 ^bcd^	12.1 ^bcd^	12.2 ^cde^											
	3	12.3 ^def^	12.0 ^bcd^	12.0 ^bcd^											
	4	11.9 ^bc^	11.2 ^a^	11.8 ^ab^											
Hb O_2_ (%)		69.3 ^a^	72.3 ^ab^	75.7 ^b^	1.30										
	1					64.5 ^ab^	59.9 ^a^	67.1 ^bc^	64.7 ^ab^	2.30					
	2					69.6 ^bcd^	72.1 ^cd^	73.6 ^def^	74.9 ^defg^						
	3					72.8 ^cde^	72.8 ^cde^	69.8 ^bcd^	80.2 ^fg^						
	4					78.3 ^efg^	79.3 ^fg^	80.8 ^g^	78.4 ^efg^						
O_2_ (mmol/L)		5.07 ^a^	5.20 ^a^	5.53 ^b^	0.114	5.32	5.08	5.19	5.47	0.130	4.92 ^a^	5.26 ^b^	5.36 ^bc^	5.52 ^c^	0.093
PCV (%)		31.8	31.6	32.7	0.44	33.1 ^b^	31.4 ^a^	31.3^a^	32.4 ^ab^	0.51	33.9 ^b^	31.6 ^a^	31.6 ^a^	31.1 ^a^	0.32

^1^ Samples were collected at 13:00 h at the end of week 2, 4, 6, and 8, which were the second weeks in each 2 weeks period with increasing heat load index. Target daytime/nighttime heat load indexes were 70/70, 85/70, 90/77, and 95/81 in period 1, 2, 3, and 4, respectively. ^2^ DOR, Dorper; KAT, Katahdin; STC, St. Croix. ^3^ MW, Midwest; NW, Northwest; SE, Southeast; TX, central Texas. ^4^ Target daytime/nighttime heat load indexes were 70/70, 85/70, 90/77, and 95/81 in period 1, 2, 3, and 4, respectively. ^5^ Hb, hemoglobin; O_2_, oxygen; PCV, packed cell volume. ^a,b,c,d,e,f,g^ Means within grouping without a common superscript letter differ (*p* < 0.05).

**Table 4 animals-12-02273-t004:** *p* values for effects of breed, region, and period with different heat load indexes on serum concentrations of glucose, lactate, urea nitrogen, creatinine, total protein, albumin, triglycerides, cholesterol, cortisol, thyroxine, and heat shock protein 70 in hair sheep ^1^.

	Source of Variation ^2^								
Item ^3^	Set	Age	BRD	REG	BR×REG	PRD	BRD×PRD	REG×PRD	BRD×REG×PRD
GLC (mg/dL)	<0.001	0.001	0.031	0.214	0.295	0.853	0.970	0.844	0.799
LAC (mg/dL)	<0.001	0.213	0.646	0.044	0.934	0.005	0.721	0.911	0.241
UN (mg/dL)	<0.001	0.256	<0.001	0.106	0.021	0.204	0.084	0.741	0.902
CRT (mg/dL)	<0.001	0.820	<0.001	0.479	0.092	<0.001	0.950	0.609	0.602
TP (g/l)	<0.001	0.257	0.016	0.292	0.492	0.016	0.526	0.122	0.918
ALB (g/l)	<0.001	0.040	0.061	0.945	0.175	0.067	0.257	0.410	0.909
TG (mg/dL)	<0.001	0.172	<0.001	0.993	0.006	0.455	0.347	0.725	0.872
CHL (mg/dL)	<0.001	0.347	0.090	0.138	0.922	0.032	0.877	0.303	0.636
COR (ng/mL)	0.032	0.311	0.003	0.781	0.384	0.020	0.854	0.448	0.516
THY (μg/dL)	0.168	0.186	0.399	0.942	0.134	<0.001	0.063	0.643	0.060
HSP (ng/mL)	0.001	0.698	0.341	0.608	0.548	0.030	0.792	0.658	0.938

^1^ Samples were collected at 13:00 h at the end of week 2, 4, 6, and 8, which were the second weeks in each 2 weeks period with increasing heat load index. Target daytime/nighttime heat load indexes were 70/70, 85/70, 90/77, and 95/81 in period 1, 2, 3, and 4, respectively. Cortisol, thyroxine, and heat shock protein 70 concentrations were determined in samples collected at the end of week 2 and 8. ^2^ BRD, breed; REG, region; PRD, period. ^3^ GLC, glucose; LAC, lactate; UN, urea nitrogen; CRT, creatinine; TP, total protein; ALB, albumin; TG, triglycerides; CHL, cholesterol; COR, cortisol; THY, thyroxine; HSP, heat shock protein 70.

**Table 5 animals-12-02273-t005:** Effects of breed, region, and period with different heat load indexes on serum concentrations of glucose, lactate, urea nitrogen, creatinine, total protein, albumin, triglycerides, cholesterol, cortisol, thyroxin, and heat shock protein 70 in hair sheep ^1^.

		Breed ^2^				Region ^3^					Period ^4^				
Item ^5^	Breed	DOR	KAT	STC	SEM	MW	NW	SE	TX	SEM	1	2	3	4	SEM
GLC		50.0 ^a^	52.6 ^b^	52.1 ^b^	0.76	50.4	52.8	51.2	51.8	0.87	52.0	51.4	51.1	51.6	0.75
LAC		26.8	25.6	26.4	1.01	28.4 ^a^	27.3 ^a^	25.3 ^ab^	24.0 ^b^	1.15	27.9 ^b^	25.3 ^ab^	27.8 ^b^	24.0 ^a^	0.99
UN											17.9	18.5	18.1	17.8	0.30
	DOR					17.4 ^abc^	18.0 ^bcd^	16.7^ab^	17.2 ^abc^	0.69					
	KAT					17.7 ^abc^	15.9 ^a^	18.9^cd^	17.4 ^abc^						
	STC					19.1 ^cde^	19.9 ^de^	21.1^e^	17.9 ^bc^						
CRT		0.99 ^c^	0.86 ^b^	0.80 ^a^	0.015	0.88	0.91	0.88	0.87	0.017	0.84 ^a^	0.91 ^c^	0.87 ^b^	0.92 ^c^	0.013
TP		6.50 ^a^	6.68 ^ab^	6.95 ^b^	0.017	6.63	6.91	6.70	6.60	0.122	6.94 ^b^	6.66 ^a^	6.60 ^a^	6.65 ^a^	0.094
ALB		2.46	2.55	2.56	0.032	2.51	2.54	2.53	2.52	0.038	2.60	2.50	2.50	2.51	0.033
TG											26.8	26.4	27.1	27.5	0.68
	DOR					28.6 ^cde^	25.4 ^abc^	27.3 ^bcd^	31.9 ^de^	1.78					
	KAT					27.8 ^bcd^	33.5 ^e^	29.2 ^cde^	25.3 ^abc^						
	STC					24.5 ^abc^	22.6 ^a^	23.7 ^abc^	23.4 ^ab^						
CHL		55.9	61.1	57.5	1.74	58.2	54.3	59.9	60.1	1.99	60.2 ^b^	56.5 ^a^	58.3 ^ab^	57.6 ^a^	1.26
COR ^6^		6.30 ^a^	8.79 ^b^	6.22 ^a^	0.596	7.08	7.10	6.60	7.64	0.684	7.62 ^b^			6.59 ^a^	0.404
THY ^6^		5.29	5.65	5.31	0.215	5.52	5.39	5.44	5.31	0.246	5.83 ^b^			5.00 ^a^	0.140
HSP ^6^		147	136	138	5.6	138	137	142	148	6.5	136 ^a^			146 ^b^	4.0

^1^ Samples were collected at 13:00 h at the end of week 2, 4, 6, and 8, which were the second weeks in each 2 weeks period with increasing heat load index. Target daytime/nighttime heat load indexes were 70/70, 85/70, 90/77, and 95/81 in period 1, 2, 3, and 4, respectively. ^2^ DOR, Dorper; KAT, Katahdin; STC, St. Croix. ^3^ MW, Midwest; NW, Northwest; SE, Southeast; TX, central Texas. ^4^ Target daytime/night time heat load indexes were 70/70, 85/70, 90/77, and 95/81 in period 1, 2, 3, and 4, respectively. ^5^ GLC, glucose (mg/dL); LAC, lactate (mg/dL); UN, urea nitrogen (mg/dL); CRT, creatinine (mg/dL); TP, total protein (g/l); ALB, albumin (g/l); TG, triglycerides (mg/dL); CHL, cholesterol (mg/dL); COR, cortisol (ng/mL); THY, thyroxine (μg/dL); HSP, heat shock protein 70 (ng/mL). ^6^ Cortisol, thyroxine, and heat shock protein 70 concentrations were determined in blood samples collected at the end of week 2 and 8. ^a,b,c,d,e^ Means within grouping without a common superscript letter differ (*p* < 0.05).

**Table 6 animals-12-02273-t006:** *p* values for effects of breed, region, and week within period 4 on hemoglobin concentration and oxygen saturation, blood O_2_ concentration, packed cell volume, and serum concentrations of glucose, lactate, urea nitrogen, creatinine, total protein, albumin, triglycerides, and cholesterol in hair sheep ^1^.

	Source of Variation ^2^								
Item ^3^	Set	Age	BRD	REG	BRD × REG	WK	BRD × WK	REG × WK	BRD×REG × WK
Hb (g/dL)	<0.001	0.720	0.350	0.394	0.316	<0.001	0.007	0.580	0.568
Hb O_2_ (%)	<0.001	0.135	0.046	0.968	0.304	0.062	0.851	0.959	0.597
O_2_ (mmol/L)	<0.001	0.153	0.362	0.685	0.097	0.632	0.424	0.807	0.473
PCV (%)	<0.001	0.619	0.326	0.123	0.461	<0.001	0.021	0.584	0.936
GLC (mg/dL)	<0.001	0.022	0.067	0.126	0.695	0.261	0.687	0.231	0.675
LAC (mg/dL)	<0.001	0.733	0.972	0.111	0.601	0.013	0.803	0.816	0.154
UN (mg/dL)	<0.001	0.315	<0.001	0.502	0.018	0.172	0.762	0.144	0.167
CRT (mg/dL)	<0.001	0.723	<0.001	0.710	0.125	0.176	0.702	0.024	0.116
TP (g/l)	<0.001	0.354	0.005	0.700	0.633	0.298	0.967	0.094	0.246
ALB (g/l)	<0.001	0.128	0.013	0.970	0.240	0.304	0.975	0.072	0.169
TG (mg/dL)	<0.001	0.542	<0.001	0.719	0.021	0.630	0.445	0.866	0.814
CHL (mg/dL)	<0.001	0.560	0.139	0.207	0.907	0.797	0.810	0.456	0.456

^1^ Samples were collected at 13:00 h at the end of week 7 and 8, which were the first and second weeks, respectively, of period 4 with target daytime/nighttime heat load indexes of 95/81. ^2^ BRD, breed; REG, region; WK, week. ^3^ Hb, hemoglobin; PCV, packed cell volume; GLC, glucose; LAC, lactate; UN, urea nitrogen; CRT, creatinine; TP, total protein; ALB, albumin; TG, triglycerides; CHL, cholesterol.

**Table 7 animals-12-02273-t007:** Effects of breed, region, and week within period 4 on hemoglobin concentration and oxygen saturation, blood O_2_ concentration, packed cell volume, and serum concentrations of glucose, lactate, urea nitrogen, creatinine, total protein, albumin, triglycerides, and cholesterol in hair sheep ^1^.

			Breed ^2^				Region ^3^					Week ^4^		
Item ^5^	Breed	Week	DOR	KAT	STC	SEM	MW	NW	SE	TX	SEM	7	8	SEM
Hb		7	12.3 ^b^	12.2 ^b^	12.1 ^b^	0.21	12.1	11.7	11.8	12.1	0.23			
		8	11.9 ^b^	11.2 ^a^	11.9 ^b^									
Hb O_2_			74.9 ^a^	79.1 ^b^	79.4 ^b^	1.45	78.0	77.5	78.4	77.3	1.66	76.4	79.2	1.10
O_2_			5.42	5.53	5.68	0.126	5.64	5.42	5.50	5.60	0.145	5.57	5.52	0.089
PCV		7	32.5 ^c^	32.3 ^bc^	32.4 ^bc^	0.55	32.5	31.2	30.9	32.3	0.57			
		8	31.0 ^ab^	30.1 ^a^	32.1 ^bc^									
GLC			49.2	51.3	52.6	1.01	50.9	53.1	51.0	49.1	1.15	50.5	51.5	0.73
LAC			25.6	25.7	26.0	1.32	28.0	27.2	24.2	23.6	1.50	27.5^b^	24.0^a^	1.02
UN	DOR						16.5 ^ab^	17.4 ^abc^	15.7 ^ab^	17.1 ^ab^	0.86	17.4	17.8	0.28
	KAT						16.9 ^ab^	15.2 ^a^	17.5 ^bc^	17.2 ^abc^				
	STC						19.6 ^cd^	20.0 ^cd^	21.3 ^d^	17.1 ^ab^				
CRT			1.01 ^b^	0.88 ^a^	0.83 ^a^	0.020								
		7					0.86 ^a^	0.94 ^c^	0.92 ^abc^	0.88 ^ab^	0.026			
		8					0.92 ^bc^	0.91 ^abc^	0.90 ^abc^	0.93 ^c^				
TP			6.34 ^a^	6.49 ^a^	6.96 ^b^	0.131	6.58	6.74	6.56	6.50	0.150	6.54	6.65	0.092
ALB			2.41 ^a^	2.48 ^ab^	2.58 ^b^	0.039	2.47	2.49	2.50	2.49	0.045	2.47	2.51	0.030
TG	DOR						29.1 ^bcd^	26.7 ^abc^	26.3 ^abc^	33.2 ^de^	2.24	27.2	27.6	0.73
	KAT						31.1 ^cd^	35.1 ^e^	28.9 ^bcd^	24.7 ^abc^				
	STC						24.1 ^ab^	22.7 ^a^	23.8 ^ab^	23.4 ^ab^				
CHL			55.4	60.8	56.6	2.05	58.1	53.5	59.4	59.6	2.35	57.5	57.8	1.29

^1^ Samples were collected at 13:00 h at the end of week 7 and 8, which were the first and second weeks, respectively, of period 4 with target daytime/nighttime heat load indexes of 95/81. ^2^ DOR, Dorper; KAT, Katahdin; STC, St. Croix. ^3^ MW, Midwest; NW, Northwest; SE, Southeast; TX, central Texas. ^4^ week 7 and 8 were the first and second of period 4 with a target daytime and nighttime heat load indexes of 95 and 81, respectively. ^5^ Hb, hemoglobin (g/dL); O_2_ (mmol/L); PCV, packed cell volume (%); GLC, glucose (mg/dL); LAC, lactate (mg/dL); UN, urea nitrogen (mg/dL); CRT, creatinine (mg/dL); TP, total protein (g/l); ALB, albumin (g/l); TG, triglycerides (mg/dL); CHL, cholesterol (mg/dL). ^a,b,c,d^ Means within grouping without a common superscript letter differ (*p* < 0.05).

## Data Availability

Mean data are presented in tables.

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
