# Peer review of "Effects of High Heat Load Conditions on Blood Constituent Concentrations in Dorper, Katahdin, and St. Croix Sheep from Different Regions of the USA"

_animals, 2022, doi:10.3390/ani12172273_

Round 1

Reviewer 1 Report

Line 131: For the reader's better understanding, the authors are advised to report the description of the HLI calculation.

For better understanding of the p value tables and to streamline the very large tables, it is recommended that only the P value of significant parameters be reported

Author Response

The changes in the revised manuscript have been highlighted in yellow color and the responses of the reviewers’ comments/suggestions are indicated as “Authors’ response” in bold fonts.

Reviewer #1

Line 131: For the reader's better understanding, the authors are advised to report the description of the HLI calculation.

Authors’ response: The formula for the calculation is now provided.

For better understanding of the p value tables and to streamline the very large tables, it is recommended that only the P value of significant parameters be reported

Authors’ response: P-values are reported in separate tables, which will be useful for the readers to understand the main effects and their interactions for this complex experimental design. The mean values of the main effects are presented in separate tables when only main effects are significant with superscripts letters at p< 0.05 or are not significant. But mean values for the interaction effects are presented when interactions effects were significant. Readers can quickly cross-check the corresponding p-values from the p-value tables and could comprehend them quickly. We hope the Reviewer can understand our points.

Reviewer 2 Report

Manuscirpt ID: animals-1879618

Title: Effects of high heat load conditions on blood constituent concentrations in Dorper, Katahdin, and St. Croix sheep from different regions of the USA

Heat stress is becoming an important constraint for sheep productivity in the world. In this manuscript, the authors investigate the variation of some blood constituents affected by increasing heat load conditions in three hair sheep breeds from four regions. The authors found that the levels of detected blood constituents were not highly related to resilience to high HLI. The results are interesting and might be useful for sheep industry.

General comments

In the introduction section, it is better to explain why the heat load index was selected to measure the heat stress.

In results section, Several tables are too complicated, it is better to show the most import results ,e.g., significant results, in these tables and others, e.g., not significant results, could document in additional files.

Specific comments

Line58, Please provide the inventory of these three breeds in the .U.S.

Line131, Please briefly described the formula to calculate heat load index

Lines122-124, In Figure1, please add sample size of each breed in each region.

Line156 Please indicate SAS Version

Line 156-157 Mixed Effects Model usually including Fixed Effects and Random Effects, which variable(s) was (were) regarded as random effect?

Line 433, please add page number

Lines 449,458 , doi was added in this reference

Author Response

Reviewer #2

Heat stress is becoming an important constraint for sheep productivity in the world. In this manuscript, the authors investigate the variation of some blood constituents affected by increasing heat load conditions in three hair sheep breeds from four regions. The authors found that the levels of detected blood constituents were not highly related to resilience to high HLI. The results are interesting and might be useful for sheep industry.

Authors’ response: Thank you for the good suggestions.

General comments

In the introduction section, it is better to explain why the heat load index was selected to measure the heat stress.

Authors’ response: We have introduced a paragraph to explain HLI as a better measure of effective heat stress in livestock.

In results section, Several tables are too complicated, it is better to show the most important results ,e.g., significant results, in these tables and others, e.g., not significant results, could document in additional files.

Authors’ response: Authors’ response: p-values are reported in separate tables, which will be useful for the readers to understand the main effects and their interactions for this complex experimental design. The mean values of the main effects are presented in separate tables when only main effects are significant with superscripts letters at P< 0.05 or are not significant. But mean values for the interaction effects are presented when interactions effects were significant. Readers can quickly cross-check the corresponding p-values from the p-value tables and could comprehend them quickly. We hope the Reviewer can understand our points.

Specific comments

Line 58, Please provide the inventory of these three breeds in the .U.S.

Authors’ response: There is no official US Government statistics on breed inventory, but breed association registration records indicate that Dorper and Katahdin hair sheep breed registrations have increased over 8000 and 10000 in 2017 from around 2000 and 4000 in 2000, respectively. We have mentioned this information in the revised manuscript.

Line131, Please briefly described the formula to calculate heat load index

Authors’ response: The formula for the calculation has been provided now.

Lines122-124, In Figure1, please add sample size of each breed in each region.

Authors’ response: Sample sizes of each breed from each region are provided now (in parenthesis).

Line156 Please indicate SAS Version

Authors’ response: The version was listed in the reference, but now has also been included in the text as suggested by the reviewer.

Line 156-157 Mixed Effects Model usually including Fixed Effects and Random Effects, which variable(s) was (were) regarded as random effect?

Authors’ response: A sentence about the random effect has been inserted.

Line 433, please add page number

Authors’ response: Perhaps the line indicated could be 443 rather than 433. This journal now provides an article ID number but not page numbers.

Lines 449,458 , doi was added in this reference

Authors’ response: doi numbers have been removed now.
